# Alexithymia Is Linked with a Negative Bias for Past and Current Events in Healthy Humans

**DOI:** 10.3390/ijerph18136696

**Published:** 2021-06-22

**Authors:** Silvia Barchetta, Gabriella Martino, Giuseppe Craparo, Mohammad A. Salehinejad, Michael A. Nitsche, Carmelo M. Vicario

**Affiliations:** 1Department of Cognitive Sciences, Psicologiche, Pedagogiche e Degli Studi Culturali, Università di Messina, 98122 Messina, Italy; s.barchetta95@gmail.it; 2Department of Clinical and Experimental Medicine, University of Messina, 98122 Messina, Italy; gabriella.martino@unime.it; 3Faculty of Human and Social Sciences, UKE-Kore University of Enna, Cittadella Universitaria, 94100 Enna, Italy; giuseppe.craparo@unikore.it; 4Department of Psychology and Neurosciences, Leibniz Research Centre for Working Environment and Human Factors, 44139 Dortmund, Germany; salehinejad@ifado.de (M.A.S.); nitsche@ifado.de (M.A.N.); 5Department of Neurology, University Medical Hospital Bergmannsheil, Bürkle de La Camp-Platz 1, 44789 Bochum, Germany

**Keywords:** time perspective, alexithymia, TAS-20, ZTPI, past-negative, present-fatalistic

## Abstract

Although research provides a rich literature about the influence of emotional states on temporal cognition, evidence about the influence of the style of emotion processing, as a personality trait, on temporal cognition is extremely limited. We provide a novel contribution to the field by exploring the relationship between difficulties of identifying and describing feelings and emotions (alexithymia) and time perspective. One hundred and forty-two healthy participants completed an online version of the TAS-20 scale, which measures alexithymia, and the Zimbardo Time Perspective Inventory, which monitors individual differences in time-orientation regarding the past, present, and future. The results show greater attention to past negative aspects in participants whose TAS-20 score was indicating borderline or manifest alexithymia, as compared to non-alexithymic individuals. Moreover, the higher the TAS-20 score, the higher the tendency was to focus on negative aspects of the past and interpret the present fatalistically. These results suggest that difficulties in identifying and describing feelings and emotions are associated with a negative bias for past and present events. Theoretical and clinical implications of this finding are discussed.

## 1. Introduction

Our ability to represent time is the result of an intricate interaction between several variables, which are provided by the surrounding environment (e.g., space and quantity information) [1,2,3] or related to our internal states, individual variables (e.g., executive functions, arousal, interoception, aging, handedness, body weight [4,5,6,7,8,9] and emotions, as shown by a rich literature in the field [10,11,12,13,14,15,16] have shown that the duration of emotional sounds is perceived to last longer than judgement of neutral sounds. Similar time distortions were documented in the perception of facial stimuli. Droit-Volet et al. [12] have shown that the perceived duration of the presentation of angry faces was judged to last longer than the duration of neutral face presentation. Finally, time perception can be influenced by mood manipulation. Droit-Volet et al. [17] found that the duration of a neutral stimulus was judged to be longer after viewing a frightening movie that increased the emotion of fear as compared to the presentation of the same stimulus preceding exposure to this movie.

Although research on emotions and temporal processing offers a rich literature, investigations on how the personality trait emotion processing style affects time perception is extremely limited. An exception is the study by Stolarski et al. [18] showing that more temporally balanced individuals show higher levels of emotional intelligence. Moreover, Wittmann et al. [19] found that a balanced time perspective, defined by Lewin [20] (p. 75) as “the totality of individual’s view of his psychological future and psychological past existing at a given time” and measured via the Zimbardo Time Perspective Inventory (ZTPI, [21]), is linked to higher acceptance of one’s own emotions, as measured via the Scale for Experiencing Emotions [22]. Balanced time perspective is a specific pattern of Time Perspective allowing one to effectively switch between time orientations in response to situational demands [23]. Moreover, enhanced emotion regulation experience is related to reduced deviation from the ideal time perspective. All these studies deliver evidence for the beneficial effect of emotional intelligence and regulation to the experience of time. However, no research has investigated whether and how a disturbance of affective-emotional processing is associated with the experience of time.

We addressed this aspect by investigating the temporal perspective in alexithymia. Alexithymia has been described as a stable personality characteristic, or trait [24,25] characterized by a disturbance of affective-emotional processing that causes difficulties in identifying and describing feelings and emotions verbally [26]. Individuals with alexithymia also have difficulties discerning between feelings and somatic sensations that accompany emotional arousal, which can cause somatic illness [27,28,29,30,31]. Furthermore, alexithymia is characterized by externally oriented thinking, reduced daydreaming, and impaired symbolic activity [32]. For all these reasons, alexithymia is ideal to assess the role of emotion appraisal as a personality trait, i.e., the habitual tendency of individuals with regard to emotion processing, on temporal perception in the absence of experimental manipulation involving exposure to emotional stimuli or conditions with the aim of influencing actual mood (i.e., state emotion condition). Moreover, our study contributes to the understanding of time perspective as a relevant associate of psychopathology [33] with respect to alexithymia, in the absence of concomitant psychiatric disorders that might confound the effect of this trait on the time perspective.

Based on evidence of emotion dysregulation in alexithymia (e.g., [34], one may hypothesize that difficulties in describing/interpreting personal and others’ emotions, as measured via the TAS-20, correlate with the temporal perspective of individuals. In line with the work of Wittmann et al. [19], which links a balanced time perspective with the ability to accept and regulate one’s own emotions, one may expect to detect a negative time perspective bias (i.e., a generally negative, aversive view of past and current events and the concept that fate determines most personal life events) in individuals with alexithymia.

## 2. Participants

142 healthy subjects took part in this study (105 females and 37 males, age range between 19 and 39 years). The sample included 79 students, 52 employees, and 11 unemployed participants. Participants were divided in three groups—alexithymic subjects (*n* = 20), borderline (*n* = 26), and non-alexithymic subjects (*n* = 96), according to respective TAS-20 scores. More details on how groups were created are provided in the Materials and Measures section.

The average age was 24.88 years, ± 4.31 SD (age range was 19–39). For demographic data, a one-way ANOVA showed no significant difference in age across the three groups [F(2, 139) = 2.25, *p* = 0.289, η_p_^2^ = 0.017]. Neither did a difference emerge for gender [χ^2^ = 1.641, *p* = 0.445], or occupation [χ^2^ = 8.767, *p* = 0.067]. All participants gave their written informed consent prior to inclusion in the study and were naïve to its purpose. The data were anonymously collected. The study was approved by the local ethics board and was conducted in agreement with the principles of the Declaration of Helsinki.

## 3. Materials and Measures

### 3.1. TAS-20 (20 Item-Toronto Alexythymia Scale)

The TAS-20 is a self-reported scale used to measure alexithymia. It is composed of three subscales: (a) Difficulty in identifying feelings (DIF); (b) Difficulty in describing feelings (DDF); and (c) External oriented thought (EOT). Bagby et al. [35] have proposed three cut-off scores for the classification of individuals: alexithymic subjects (≥61), borderline (score range between 51 to 60), and non-alexithymic subjects (≤50). In this study, the Italian version of the Toronto Alexithymia Scale (20 items, TAS-20) was used, which was validated by Bressi et al. ([36], Cronbach’s alpha 0.75).

### 3.2. Zimbardo Time Perspective Inventory (ZTPI)

The Zimbardo Time Perspective Inventory (ZTPI) [21] was established to detect individual differences in time-orientation regarding different aspects of the past, present, and future. The ZTPI is composed of 56 items, which represent a combination of five main domains of the temporal perspective, for which participants have to indicate their level of agreement on a 5 point Likert scale: (1) past negative (PN), a generally negative, aversive view of the past; (2) past positive (PP), a warm, sentimental, positive attitude towards the past; (3) present fatalistic (PF), the concept that fate determines most personal life events; (4) present hedonistic (PE), which reflects a hedonistic, enjoyment- and pleasure-centered, risk-taking ‘devil may care’ attitude towards time and life; (5) Future (F), which measures general future orientation, planning for the achievement of future goals. According to Zimbardo & Sword [37], the “ideal temporal perspective” can be summarized by a low orientation towards negative aspects of the past, and fatalistic view of the present, and high orientation towards positive aspects of the past, as well as a hedonistic view of the present and future.

### 3.3. Procedure

All participants were contacted via social media (Facebook, Instagram). Next, they were asked to complete the test via Google forms. Before completing the TAS-20 and the ZTPI, participants were informed about the study procedures, and asked to read and accept a consent form to take part in the study.

### 3.4. Data Analysis

Statistical analysis was carried out by STATISTICA software, version 8.0 (Stat soft Inc., Tulsa, OK, USA). First, we performed the Shapiro–Wilk test of normality to establish whether parametric or not parametric analyses were to be applied. As the results of the Shapiro–Wilk showed a non-normal distribution (W = 0.970, *p* = 0.004), no parametric statistics were applied. Data were entered in a Kruskal–Wallis ANOVA to identify differences between the three groups (alexithymic, *n* = 20; Borderline, *n* = 23; Non-alexithymic, *n* = 99, according to TAS-20 scores) with regard to their temporal perspective measured via the ZTPI. Spearman correlation analyses were performed to investigate the association between the three sub-scales of the TAS-20 and the five subscales of ZTPI scores. The respective *p*-level was corrected for multiple comparisons. Therefore, the *p* level of significance was set to ≤0.003 (i.e., 0.05/15).

## 4. Results

The ANOVA showed a significant difference between groups for the past-negative subscale [H(2, 142) = 28.06, *p* < 0.001, η^2^ = 0.19]. Post hoc tests revealed a lower score for the non-alexithymic group (Sum rank = 5654.5), compared to borderline (Sum rank = 2478.5, *p* < 0.001, Cohen’s d = 0.9936) and manifest alexithymia groups (Sum rank = 2020, *p* < 0.001, Cohen’s d = 1.1679) (Figure 1A). We also found a significant difference in the present fatalistic subscale [H(2, 142) = 15.49, *p* < 0.001, η^2^ = 0.09]. Post hoc tests revealed a lower score of the non-alexithymic group (Sum rank = 6005.5), as compared to the borderline group (Sum rank = 2509.5, *p* < 0.001, Cohen’s d = 0.8568. Figure 1B). No further significant differences between the three groups emerged for the other ZTPI subscales.

Significant correlations were revealed when plotting scores of the three TAS-20 subscales with the scores of the five ZTPI subscales. We found a positive correlation between ‘difficulty in describing feelings’ and ‘past negative’ scores. Moreover, the ‘difficulty in identifying feelings’ scores positively correlated with ‘past negative’ and ‘present fatalistic’ scores. Table 1 provides details of all correlations.

## 5. Discussion

Several studies have shown that emotions affect time processing [13,38,39,40]. Moreover, a balanced time perspective is linked to an appropriate regulation of one’s own emotions [19] and emotional intelligence [18]. In line with this evidence, our data suggest that the trait-like individual difficulty in describing/interpreting personal and others’ emotions, as found in alexithymia and assessed via the TAS-20, is associated with a negative time perspective bias.

The results of the present study show higher scores for the past negative subscale in participants classified as borderline and manifest alexithymic, compared to participants with non-alexithymic TAS-20 scores. Moreover, we found higher scores of the present fatalistic subscale in participants classified as borderline compared to participants with non-alexithymic TAS-20 scores. No significant difference was found when comparing non-alexithymic and manifest alexithymia groups. This result can be explained by the low number of participants in the manifest alexithymia group. Finally, we found positive correlations between TAS-20 and ZTPI subscales. In particular, the higher the difficulty in describing feelings (DDF), the higher the score on the past negative subscale. Moreover, the higher the difficulty in identifying feelings (DIF), the higher the scores on the past negative and the present fatalistic subscales. These data suggest that both DDF and DIF scores are associated with the severity of the bias for negative past events, while only DIF scores are associated with the bias for a fatalistic vision/concept of the present.

These results are in line with the suggestion that people with a more severe alexithymia trait are affected by maladaptive rumination [41], defined as negative thinking about the past [42], but also about the present. Moreover, we show that the higher the TAS-20 score, the greater the tendency to interpret the present fatalistically. This may be interpreted as evidence that people with more severe alexithymia tend to perceive themselves as powerless, interpreting their present as externally predetermined or ruled by fate.

In summary, our study shows that in people with borderline and manifest alexithymia TAS-20 scores, the temporal perspective for the past and the present is unbalanced or biased toward negative representations and memories. This, in turn, may contribute to explaining the high predisposition of alexithymic people to develop psychopathological and psychosomatic disorders [27,43].

In their seminal study, Zimbardo and Boyd [21] reported that past negative and present fatalistic perspectives are associated with depression and anxiety. More recently, Van Beek et al. [44] reported that higher scores in the present fatalistic subscale are associated with lower extraversion and higher neuroticism. Our results are in line with these reports, as alexithymia is associated with depression, anxiety, and is related to low extraversion and high neuroticism [45,46]. Since we did not explore depression, anxiety, extraversion, and neuroticism in our sample, we cannot establish whether and to what extent these variables may have influenced the time perspective in participants with borderline and manifest alexithymia TAS-20 scores. Nevertheless, we believe it is unlikely that depression and anxiety have played an important role in the time perspective pattern of the present study, as it did not include clinical populations.

From a neuroscientific perspective, our results are in accordance with findings that past negative and present fatalistic views in the temporal domain are both negatively associated with Grey Matter Volume (GMV) of the anterior insula [47], since alexithymia is also associated with reduced GMV in this region [48,49]. The suggestion that the negative time perspective bias of our borderline and manifest alexithymic participants may be linked with reduced insula GMV is also supported by evidence for the involvement of this region in time processing [50], and negative experiences such as aversion [51], disgust [52,53], and pain [54]. Moreover, there is evidence for abnormal right parietal cortex activation in alexithymia [55], another central region for temporal perspective taking [56], and time perception [57].

From a cognitive perspective, our results would be compatible with the hypothesis of higher distress and working memory deficits explaining the higher scores for past negative and present fatalistic views of borderline and manifest alexithymic participants. This hypothesis is based on previous work in the field documenting that high distress is related to higher past negative scores [58] and working memory performance is associated with present fatalistic scores [59]. Nevertheless, since we did not investigate respective cognitive processes in the present study, this remains to be investigated in future.

The evidence for a negative/fatalistic time perspective bias in alexithymia provided by the results of this study suggests some potentially important information for clinical practice. It may also be relevant to work on time perspective in order to improve emotional skills in this affective disorder by modifying negative bias for past and current events. This negative bias, which may contribute to explaining emotional dysregulation [60], may be responsible, at least in part, for the onset and maintenance of several psychosomatic and mental disorders (e.g., [60]) typically associated with alexithymia, such as anxiety and depression [46]. However, this statement does not mean that we should tone down the relevance of emotions in other areas of life, and that working directly on emotion processing would be less valuable.

Some limitations of the present study should be taken into consideration. First, the number of participants with a TAS-20 score that fell into the manifest alexithymia category (i.e., a score > 61) was relatively low. Furthermore, our sample was biased towards female participants. However, no gender differences were reported between the three samples. Additionally, the current dataset was limited to a questionnaire that explored the perception of time related to emotions. To discern this specific feature of time perception from the general ability to represent the duration of internal and external events in alexithymia, it would be useful to collect information about time perception via computer-based studies in future investigations. Moreover, based on our data we cannot exclude the influence of depression and anxiety on the results, which might be especially relevant in borderline and manifest alexithymia participants, as we did not collect these data. Finally, we did not investigate interoceptive awareness in our participants. Given the link between interoception and time processing [61], as well as between interoceptive awareness and alexithymia [62], the inclusion of this variable would have been valuable to clarify its influence on the reported time perspective pattern.

## 6. Conclusions

This work shows, for the first time, that individuals with borderline and manifest alexithymia are characterized by a high tendency to focus on negative aspects of the past and interpret the present fatalistically. This suggests that difficulties in identifying and describing feelings and emotions are associated with a negative bias for past and present events.

## Figures and Tables

**Figure 1 ijerph-18-06696-f001:**
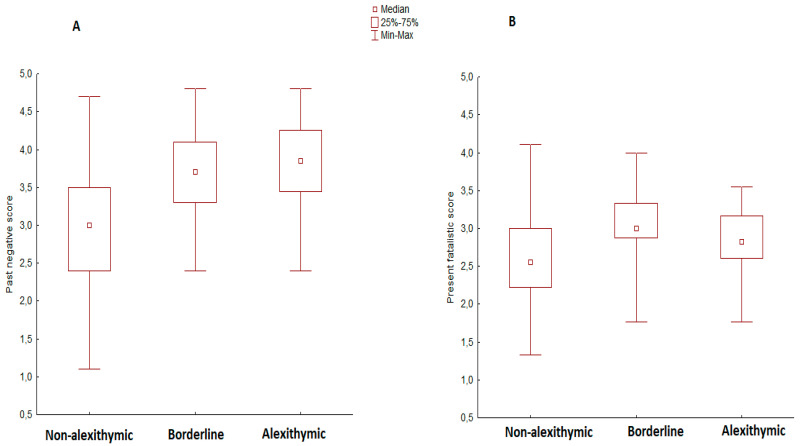
(**A**) Median scores of the three groups of participants (Non-alexithymic, borderline, alexithymic) for the past negative ZTPI subscale. (**B**) Median scores of the three groups of participants (non-alexithymic, borderline, alexithymic) for the present fatalistic ZTPI subscale. Vertical bars indicate quartile ranges.

**Table 1 ijerph-18-06696-t001:** Correlation analysis between TAS-20 scores and the scores of the ZTPI subscales. *p*-levels were corrected for multiple comparisons (the critical level of significance is ≤0.003). The confidence interval is 95%.

Difficulty in Describing Feelings (DDF)	Spearman R	Confidence Interval	*p*-Level
Past Negative	0.341	0.187|0.478	<0.001 *
Past Positive	−0.134	−0.292|0.031	0.111
Present Hedonistic	−0.023	−0.187|0.141	0.778
Present Fatalistic	0.220	0.059|0.372	0.008
Future	−0.016	−0.181|0.148	0.842
**Difficulty Identifying Feelings (DIF)**			
Past Negative	0.613	0.5|0.706	<0.001 *
Past Positive	−0.228	−0.378|−0.067	0.006
Present Hedonistic	0.167	0.003|0.323	0.046
Present Fatalistic	0.432	0.289|0.557	<0.001 *
Future	−0.230	−0.38|−0.069	0.005
**Externally Oriented Thought (EOT)**			
Past Negative	0.030	−0.135|0.194	0.719
Past Positive	0.007	−0.157|0.171	0.931
Present Hedonistic	−0.023	−0.187|0.142	0.784
Present Fatalistic	0.069	−0.096|0.231	0.410
Future	−0.234	−0.384|−0.073	0.004

* indicates significant results.

## Data Availability

The data presented in this study are available on request from the corresponding author. The data are not publicly available due to privacy reasons.

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
