# Peer review of "Alexithymia Is Linked with a Negative Bias for Past and Current Events in Healthy Humans"

_ijerph, 2021, doi:10.3390/ijerph18136696_

Round 1

Reviewer 1 Report

A very interesting correlation study. Simple, but with great application potential. The value of the analyzes will be much higher in the case of clinical groups, for example drug addicts.

Author Response

# We thank the reviewer for the positive comment. We plan to further address this research line by testing clinical populations, including drug addiction.

Reviewer 2 Report

This manuscript reports a study investigating relationship between alexithymia and individual differences in time orientation using self-report measures. Out of five measured dimensions from Zimbardo’s Time Perspective Inventory (ZTPI), the association with alexithymia score was revealed for Past negative and Present fatalistic dimensions. An increased tendency to attend to past negative aspects was shown for individuals with borderline and high alexithymia scores relative to those with low alexithymia scores. These results are interpreted in terms of heightened negative bias for past and present events being associated with difficulties to identify and describe feelings.

After reading the manuscript, my impression is that it is not ready to be published in the present form. I have the following suggestions/comments:

  • I am not entirely sure this study brings much novelty to the field. I would perhaps need more persuasion in the Introduction. This part is written as a collection of ideas, not always flowing nicely. By definition, trait alexithymia should be linked closely with disturbances in emotion processing and as such, these difficulties are usually more pronounced for negative emotions. Therefore, it shouldn’t be too surprising that individuals who perceive themselves as incapable of identifying and describing their feelings, would have pessimistic attitude toward past and present events, if we consider their possibly challenging social functioning.
  • The authors did not study temporal perspective as an ability, but only as a trait (subjective perceptions of this ability). As such, collected responses could have been affected by inaccurate introceptive capabilities of these individuals (see paragraph 3 on page 2).
  • This study neither focused on emotion regulation, nor can predict anything on the basis of employed design, so I would be more careful in formulating hypotheses (Intro, last paragraph) and perhaps also the rationale of this study.
  • Participants section seems incomplete. No information on occupation, education, and health status of the sample/contrasted groups is provided. Some data are presented in the Results (please move it up), but the description is still incomplete. The current health status is of great importance, since alexithymia can be accompanied with depression, anxiety and is frequently present in populations with neurological and/or psychiatric diseases.
  • Results section: An outline of analyses is very brief and rationale for selecting this particular approach is missing. For example, are parametric tests suitable given the nature of this dataset (ordinal level, unknown distribution)? Why was the sample divided into these particular three groups? The division seems arbitrary and resulted in quite unbalanced sizes of compared groups. Wouldn’t it be more beneficial to contrast high- vs. low- alexithymia individuals or perform data-driven analysis to explore whether there are specific types within the sample? Another option could be to look into individual facets of alexithymia and their relationship with ZTPI dimensions (this is an approach recommended by a recent review on alexithymia – see Luminet, O., Nielson, K. A., & Ridout, N. (2021). Cognitive-emotional processing in alexithymia: an integrative review. Cognition and Emotion, 1-39).
  • Related to the point above, the analysis is not specified clearly. What factors were used in the repeated ANOVA? Group (3) x Time perspective (5)?
  • I am also confused about the presented results on overall ZTPI score – what was the rationale behind this step and how can the result be interpreted?
  • Neither effect sizes for selected contrasts (Results), nor confidence intervals (Table 1) are reported. Figure 1 misses SE bars for all groups and dimensions.
  • Discussion section: I do not agree with the interpretation on page 5, line 24 on “limited capacity… to accept and learn from past mistakes and positively approach the present…”. How can the authors predict the ability to accept and learn from past mistakes on the basis of their data?
  • The same goes for the last paragraph on the page 5, suggesting working with alexithymic individuals on time perspective in order to improve their emotional skills – I believe that the deficits in emotion processing are more pervasive, affecting many areas of life and time perspective is only one of them.
  • I don’t follow the reasoning in paragraph 4, page 5
  • I believe that the paragraph discussing neuroscientific findings is not relevant for this study, since it was conducted online with very little information collected (and only via self-report). Therefore it would be more appropriate to relate these results to more suitable existing studies such as those reporting findings on cognitive biases in individuals with alexithymia, differences in emotion-information processing, or association of negative bias and other personality traits (these are just suggestions, the authors might want to include different topics).

In conclusion, the manuscript would benefit from a major revision. I recommend that the authors re-evaluate their analysis approach and include more relevant studies in the Introduction and Discussion section. I would also like to see more appropriate interpretations of the findings and generally more careful and precise formulation of ideas throughout the manuscript.

Author Response

  • I am not entirely sure this study brings much novelty to the field. I would perhaps need more persuasion in the Introduction. This part is written as a collection of ideas, not always flowing nicely. By definition, trait alexithymia should be linked closely with disturbances in emotion processing and as such, these difficulties are usually more pronounced for negative emotions. Therefore, it shouldn’t be too surprising that individuals who perceive themselves as incapable of identifying and describing their feelings, would have pessimistic attitude toward past and present events, if we consider their possibly challenging social functioning.

# Our work provides novel evidence to the extremely limited empirical literature on the influence of emotions on time perspective. The available research addressed the beneficial effect of emotional intelligence and emotional regulation on the time perspective. Our work investigated whether and how a disturbance of affective-emotional processing impacts the experience of time.

While it could be intuitively (and not surprisingly) expected that individuals who do not perceive themselves as capable of identifying and describing their feelings have a pessimistic attitude toward the past and the present, also based on challenges with respect to social functioning, the same principle, in a reversed way, should also apply (i.e., individuals who perceive themselves as capable of identifying and describing their feelings, should show a more positive attitude towards past positive and present hedonistic aspects (i.e., two ZTPI subscales), as compared to alexithymics. Moreover, in that case, a group difference would have been expected for the future subscale of the ZTPI. Our research however clarifies empirically the effects of alexithymia on the time perspective. Finally, thanks to further correlation analyses, we now clarified the specific contribution of the three TAS 20 subscales on the five dimensions of temporal perspective examined in our work. We have now modified the introduction to make all these points explicit.

  • The authors did not study temporal perspective as an ability, but only as a trait (subjective perceptions of this ability). As such, collected responses could have been affected by inaccurate introceptive capabilities of these individuals (see paragraph 3 on page 2).

We thank the reviewer for this comment. Although we cannot confirm (or exclude) the influence of interoception on the temporal perspective, as this variable was not collected, we think this is a likely possibility, according to the literature documenting an inverse relationship between interoceptive awareness and Alexithymia (e.g., Herbert et al., 2011, J of Personality). We have now included this aspect in the discussion (see pages 11 and 12).

  • This study neither focused on emotion regulation, nor can predict anything on the basis of employed design, so I would be more careful in formulating hypotheses (Intro, last paragraph) and perhaps also the rationale of this study.

While we did not focus on emotion regulation, there is evidence about difficulties in emotion regulation in alexithymia. This  is now mentioned in the respective section of the text. Moreover, our prediction is in line with what this reviewer stated in the first comment, that is, “…, it shouldn’t be too surprising that individuals who perceive themselves as incapable of identifying and describing their feelings, would have pessimistic attitude toward past and present events”. Nevertheless, we have now toned down our hypothesis.

  • Participants section seems incomplete. No information on occupation, education, and health status of the sample/contrasted groups is provided. Some data are presented in the Results (please move it up), but the description is still incomplete. The current health status is of great importance, since alexithymia can be accompanied with depression, anxiety and is frequently present in populations with neurological and/or psychiatric diseases.

We have now moved up respective data presented from the results to the participant section, as suggested. The other information asked for by this reviewer was not collected, as our participants were not selected from clinical populations. Nevertheless, we agree that this information would have enriched the study and therefore, we mention this aspect in the limitation section.   

  • Results section: An outline of analyses is very brief and rationale for selecting this particular approach is missing. For example, are parametric tests suitable given the nature of this dataset (ordinal level, unknown distribution)? Why was the sample divided into these particular three groups? The division seems arbitrary and resulted in quite unbalanced sizes of compared groups. Wouldn’t it be more beneficial to contrast high- vs. low- alexithymia individuals or perform data-driven analysis to explore whether there are specific types within the sample?

We have now included the a statistical analysis for normality. The Shapiro-Wilk test documented a non-normal data distribution. Therefore, non- parametric statistics are now applied (please refer to the data analysis and results sections of the manuscript for details).

The sampling was not arbitrary but established in accordance with the suggestions provided by the authors of the TAS-20 scale (Bagby et al., 1994), which proposed three cut-off scores for the classification of individuals: alexithymic subjects (score ≥ 61), borderline (score range between 51 to 60), and non-alexithymic subjects (≤ 50). This information is available in the text at page 5 (Materials and measures, TAS-20).

  • Another option could be to look into individual facets of alexithymia and their relationship with ZTPI dimensions (this is an approach recommended by a recent review on alexithymia – see Luminet, O., Nielson, K. A., & Ridout, N. (2021). Cognitive-emotional processing in alexithymia: an integrative review. Cognition and Emotion, 1-39).

We now provide novel correlation analyses to explore the role of individual facets of alexithymia and their relationship with ZTPI dimensions in line with this suggestion (see pages 7-9).

  • Related to the point above, the analysis is not specified clearly. What factors were used in the repeated ANOVA? Group (3) x Time perspective (5)?

The reviewer is correct in her/his statement about the ANOVA. We have modified the text to describe the statistical methods in sufficient detail, also with respect to the new analyses.

  • I am also confused about the presented results on overall ZTPI score – what was the rationale behind this step and how can the result be interpreted?

We thank the reviewer for the comment. This analysis is no longer included.

  • Neither effect sizes for selected contrasts (Results), nor confidence intervals (Table 1) are reported.

We now provide further details, as suggested (see results and table 1 sections).

  • Figure 1 misses SE bars for all groups and dimensions.

Figure 1 has been now substituted by a new figure where the  bars indicate quartile ranges of the median scores.

  • Discussion section: I do not agree with the interpretation on page 5, line 24 on “limited capacity… to accept and learn from past mistakes and positively approach the present…”. How can the authors predict the ability to accept and learn from past mistakes on the basis of their data?

 We agree, and have now removed this sentence from the text.

  • The same goes for the last paragraph on the page 5, suggesting working with alexithymic individuals on time perspective in order to improve their emotional skills – I believe that the deficits in emotion processing are more pervasive, affecting many areas of life and time perspective is only one of them.

We agree that time perspective is only one area of life. In the paragraph we proposed that working on the time perspective can improve emotional skills. This statement does not mean to exclude the relevance of emotions for other areas of life, and does not imply that working on emotion processing would be less valuable. Our intention was just to add time perspective as another relevant dimension. We have clarified this in the discussion section (see page 12).

  • I don’t follow the reasoning in paragraph 4, page 5

It would be helpful getting more details about what specifically this reviewer does not follow in the mentioned paragraph. Thank .

  • I believe that the paragraph discussing neuroscientific findings is not relevant for this study, since it was conducted online with very little information collected (and only via self-report). Therefore, it would be more appropriate to relate these results to more suitable existing studies such as those reporting findings on cognitive biases in individuals with alexithymia, differences in emotion-information processing, or association of negative bias and other personality traits (these are just suggestions, the authors might want to include different topics).

The paragraph on neural correlates has the goal to provide a perspective about the possible rationale for the abnormal time perspective pattern of borderline/alexithymia participants at the neural level. This paragraph is meant to give a background for a possible role of maladaptive rumination and fatalistic thinking in alexithymia. For this reason, we would like to keep it. On the other hand, we have now expanded the discussion on psychological processes as suggested.

In conclusion, the manuscript would benefit from a major revision. I recommend that the authors re-evaluate their analysis approach and include more relevant studies in the Introduction and Discussion section. I would also like to see more appropriate interpretations of the findings and generally more careful and precise formulation of ideas throughout the manuscript.

We have modified the manuscript to address the issues brought up, and hope that with these alterations, this reviewer is satisfied with this contribution.

Reviewer 3 Report

This is an interesting article and the authors are encouraged, as they suggest, to conduct research where information on time perception is collected through computer studies with the aim of discerning the particular feature of time perception from the general ability to represent the duration of internal and external events in alexithymia.

As strengths, it is well documented, citing research by experts in the field such as Taylor, G.J. and Bagby, R.M. as well as the latest research in the field.

As weaknesses I think that some things could be improved with respect to the participants' procedure since, as they are recruited via social media, we do not know if the participants can answer only once, or even confirm the veracity of their answers. Also, there is some imbalance in the size of the groups (Alexithymia N=20, borderline N=23; non-alexithymia N=99) which may alter the analysis of the results obtained. However, as the authors point out, this is an aspect that should be taken into consideration.

Finally, as I indicated in the review, it would be interesting and also as the authors point out, that for future research it would be useful to collect information about time perception via computer based studies.

Author Response

This is an interesting article and the authors are encouraged, as they suggest, to conduct research where information on time perception is collected through computer studies with the aim of discerning the particular feature of time perception from the general ability to represent the duration of internal and external events in alexithymia.

# We thank the reviewer for this positive comment. Because of COVID-19 Pandemic restrictions, we were not able to conduct tests which allow to discern time perception from the ability to represent the duration of events in alexithymia via computer-based protocols. We plan however to further follow this research line by adopting computer-based tests that allow to explore differences between motor and perceptual timing, as well as differences regarding the temporal scale (i.e., sub-second vs. supra-second) in alexithymia.

As strengths, it is well documented, citing research by experts in the field such as Taylor, G.J. and Bagby, R.M. as well as the latest research in the field.

We thank the reviewer for this positive comment.

As weaknesses I think that some things could be improved with respect to the participants' procedure since, as they are recruited via social media, we do not know if the participants can answer only once, or even confirm the veracity of their answers. Also, there is some imbalance in the size of the groups (Alexithymia N=20, borderline N=23; non-alexithymia N=99) which may alter the analysis of the results obtained. However, as the authors point out, this is an aspect that should be taken into consideration.

We made sure that each participant could respond only one time. Each participant was enabled to complete the questionnaires only once because the tool we used for administering the tests was Google Forms, which offers the possibility to control for this aspect. The imbalance of group sizes reflects the proportion of alexithymic individuals in the normal population. Nevertheless, we now mention different group sizes as limitation.

Finally, as I indicated in the review, it would be interesting and also as the authors point out, that for future research it would be useful to collect information about time perception via computer-based studies.

As stated above, we plan to further address this topic via computer-based timing protocols.

Round 2

Reviewer 2 Report

After reading the revision, I cannot confirm that all my comments were addressed sufficiently. In short, formulation of the ideas is rather careless throughout the manuscript and description of the results and their interpretation is relatively imprecise. Please see the specific examples below:

  • Page 2 – the authors suggest that their investigation into temporal perspective and affective processing was systematic. This is not the case.
  • Trait is not a condition.
  • The study cannot contribute to our understanding of time perspective and psychopathology if the sample was healthy. At the same time, the authors cannot exclude possibility of higher levels of depression and anxiety in their borderline and manifest alexithymia groups, because they did not collect these data (important limitation of the study).
  • Please avoid any mention of predictions – this is impossible in a correlational study
  • Statistical comparisons on demographic data are presented before explanation on division of the sample into 3 groups.
  • I asked about effect sizes for significant follow-up contrasts, not ANOVA (they are more informative).
  • I cannot see confidence intervals in the Table 1 (usually reported in the following format: e.g. 95% CI [0.2, 0.4])
  • Discussion – again, DDF and DIF scores did not predict anything
  • Rumination is not a condition.
  • I also noticed that the contrast for present fatalistic scale between non-alexythimic and manifest alexithymia group was not significant. How do the authors interpret this unexpected result?
  • The paragraphs about grey matter volume and working memory are not relevant for this study.

Author Response

After reading the revision, I cannot confirm that all my comments were addressed sufficiently. In short, formulation of the ideas is rather careless throughout the manuscript and description of the results and their interpretation is relatively imprecise. Please see the specific examples below:

We thank the reviewer for this comment, and for offering a further opportunity to improve our ms.

  • Page 2 – the authors suggest that their investigation into temporal perspective and affective processing was systematic. This is not the case.

We agree, since this study did not include interventions. We have thus now removed the term “systematic”. This is now the sentence provided at page 2: “However, no research has investigated whether and how a disturbance of affective-emotional processing is associated with the experience of time”.

  • Trait is not a condition.

We agree and have now rephrased the sentence accordingly (i.e., page 2, “For all these reasons, alexithymia is ideal to assess the role of emotion appraisal as a personality trait”)

  • The study cannot contribute to our understanding of time perspective and psychopathology if the sample was healthy. At the same time, the authors cannot exclude possibility of higher levels of depression and anxiety in their borderline and manifest alexithymia groups, because they did not collect these data (important limitation of the study).

We agree, and now mention this aspect in the limitation section (page 7).

  • Please avoid any mention of predictions – this is impossible in a correlational study

We agree, and removed the term prediction from the whole text.

  • Statistical comparisons on demographic data are presented before explanation on division of the sample into 3 groups.

We thank the reviewer for this suggestion. We now provide a short explanation on the grouping of participants before statistical comparisons of demopgraphic data (page 2).

  • I asked about effect sizes for significant follow-up contrasts, not ANOVA (they are more informative).

Many thanks for this clarification. We now provide Cohen’s d for respective significant contrasts.

  • I cannot see confidence intervals in the Table 1 (usually reported in the following format: e.g. 95% CI [0.2, 0.4])

This was now included in the table 1.

  • Discussion – again, DDF and DIF scores did not predict anything

We agree, and removed the term prediction.

  • Rumination is not a condition.

We agree, and removed the term “condition” from text (page 5).

  • I also noticed that the contrast for present fatalistic scale between non-alexythimic and manifest alexithymia group was not significant. How do the authors interpret this unexpected result?

The absence of a difference between the non-alexythimic and manifest alexithymia group in the present fatalistic scale might be due to the low number of participants in the manifest alexithymia group, and thus limited power of the present study.

  • The paragraphs about grey matter volume and working memory are not relevant for this study.

We did not remove this paragraph on neural correlates, and working memory, since we still think that these sections are important for the discussion of the results. In the previous comment of this reviewer, this removal was not mandatory (“these are just suggestions, the authors might want to include different topics”). Unless this reviewer insists on removal of this content, we  would prefer to keep it, because we think it is relevant.